# Unsymmetrical polysulfidation via designed bilateral disulfurating reagents

Jiahui Xue[1] & Xuefeng Jiang [1,2,3] ✉

Sulfur-sulfur motifs widely occur in vital function and drug design, which yearns for polysulfide construction in an efficient manner. However, it is a great challenge to install desired functional groups on both sides of sulfur-sulfur bonds at liberty. Herein, we designed a mesocyclic bilateral disulfurating reagent for sequential assembly and modular installation of polysulfides. Based on S-O bond dissociation energy imparity (mesocyclic compared to linear imparity is at least 5.34 kcal mol$^{-1}$ higher), diverse types of functional molecules can be bridged via sulfur-sulfur bonds distinctly. With these stable reagents, excellent reactivities with nucleophiles including C, N and S are comprehensively demonstrated, sequentially installing on both sides of sulfur-sulfur motif with various substituents to afford six species of unsymmetrical polysulfides including di-, tri- and even tetra-sulfides. Life-related molecules, natural products and pharmaceuticals can be successively cross-linked with sulfur-sulfur bond. Remarkably, the cyclization of tri- and tetra-peptides affords 15- and 18-membered cyclic disulfide peptides with this reagent, respectively.

[1] Shanghai Key Laboratory of Green Chemistry and Chemical Process, School of Chemistry and Molecular Engineering, East China Normal University, Shanghai, China. [2] State Key Laboratory of Organometallic Chemistry, Shanghai Institute of Organic Chemistry, Chinese Academy of Sciences, Shanghai, China. [3] State Key Laboratory of Elemento-Organic Chemistry, Nankai University, Tianjin, China. ✉email: xfjiang@chem.ecnu.edu.cn

Sulfur–sulfur bond has unique and significant roles in biological, pharmaceutical, and material fields. In organism, tertiary structures of proteins are fixed and stabilized via the linkage of sulfur–sulfur bridge among secondary structures, contributing to the versatility of proteins with complex three-dimensional structure (Fig. 1a)[1,2]. Polysulfides such as trisulfides and tetrasulfides are primary $H_2S$ donors[3], signaling of which endogenous gasotransmitter occurs via persulfidation of cysteine residues (RSH) to persulfides (RSSH) in proteins[4] with the reduction of glutathione[5] (Fig. 1b). As a powerful linker, sulfur–sulfur bridges cyclized peptide drugs with higher stability, activity, and potency compared with corresponding linear ones (Fig. 1c)[6]. Given the excellent metabolism of sulfur–sulfur bond in organism, cutting-edge drug design strategies of antibody–drug conjugates (ADCs)[7–9] and small molecule-drug conjugates (SMDCs)[10–13] involved disulfur extensively, such as Mylotarg[14] and Vintafolide[15], in which sulfur–sulfur bond serves as a reversible cross-linker. Cytotoxic drug molecule can be programmatically released relying on the reduction of glutathione when delivered to the target cells (Fig. 1d)[16]. Furthermore, polysulfides also possess high-capacity potential in cathode materials for rechargeable lithium battery, among which outstanding capacity of tetrasulfides are higher than that of trisulfides (Fig. 1e)[17–19].

Despite of the great significance of disulfur, the construction of disulfide is not yet unhindered since of high reactivity from sulfur–sulfur bond[20–24]. Though both nucleophilic[25] and electrophilic[26] disulfurating reagents have been developed, free and flexible installation on both sides of sulfur–sulfur motif is still an insurmountable challenge. Based on our concept of mask effect[27], we envision that disulfurating reagents with bilateral masks will cross-link two designated functional molecules with sulfur–sulfur bond sequentially and modularly (Fig. 2a). Dialkoxydisulfide[28,29] and diaminodisulfide[30–32] have been investigated as sulfur transfer reagents since 1970s. However, unsymmetrical polysulfidation with disulfanyl motif has never been achieved owing to the sharp contradiction raised by sequential and selective cleavage of dual S-O(N) bonds. Based on preliminary calculation of an assumed disulfurating process with molecular mechanics method (MM2, Fig. 2b), energy released from the first S-O cleavage is higher than the second due to the ring tension energy

(>5.34 kcal mol$^{-1}$) when application of mesocyclic bilateral disulfide, enabling to differentiate S-O bond cleavage (for details, see the Supplementary Figs. 7–9). Herein, we show a mesocyclic bilateral disulfurating reagent for sequential assembly and modular installation of unsymmetric polysulfides.

## Results

**Syntheses of diaza-disulfides 3 and aza-trisulfides 4.** With this concept, a series of bilateral disulfurating reagents were synthesized (Fig. 3a), whose structures were further confirmed through X-ray analysis of **1f**. In order to demonstrate the ladder-type reactivities of reagents, aniline was used as a nucleophile under the assistance of $B(C_6F_5)_3$ as catalyst (Fig. 3b). As expected, linear disulfurating reagents **1a** and **1b** resulted in poor selectivities between two S-O/N bonds, bringing mixture when coupling with aniline. Cyclic diaminodisulfide **1c** refused to transfer disulfur owing to week reactivity. Cyclic disulfane **1d** and **1e** failed to generate mono-coupling product **2** owing to the decomposition of starting material. Mono-aza-disulfide **2f** was quantitatively obtained when 10-membered disulfane **1f** was employed as a disulfurating reagent (for details, see the Supplementary Table 1).

Since the first S-O cleavage was controllably realized, another nucleophile was subsequently subjected to aza-disulfide **2f** under the assistance of weak base lithium carbonate, affording unsymmetrical diaza-disulfides **3** in Fig. 4. Diverse anilines bearing electron-withdrawing and electron-donating functional groups could be cross-linked with benzyl amines, straight-chain alkylamines, diallylamine, pyridyl methylamine, tryptamine, and even amino-acid esters at liberty (**3a**–**3j**). The unsymmetrical diaza-disulfide structure of **3a** was further confirmed through X-ray analysis. Among them, compound **3d** was afforded with a yield of 55% with occurrence of the polymerization of vinyl group accelerated by $B(C_6F_5)_3$[33]. The relative low efficiency of **3l**, **3n**, and **3o** with tryptophan motif resulted from a cyclic disulfide intermediate generated from nucleophilic cyclization of 2-position of indole (for details, see the Supplementary Fig. 2). Amines with sensitive enamine structure, amino-acid esters and antibiotic sulfamethazine underwent the connection smoothly (**3k**–**3m**). Moreover, two different amino-acid esters could be cross-linked through sulfur–sulfur bond straightforward (**3n** and

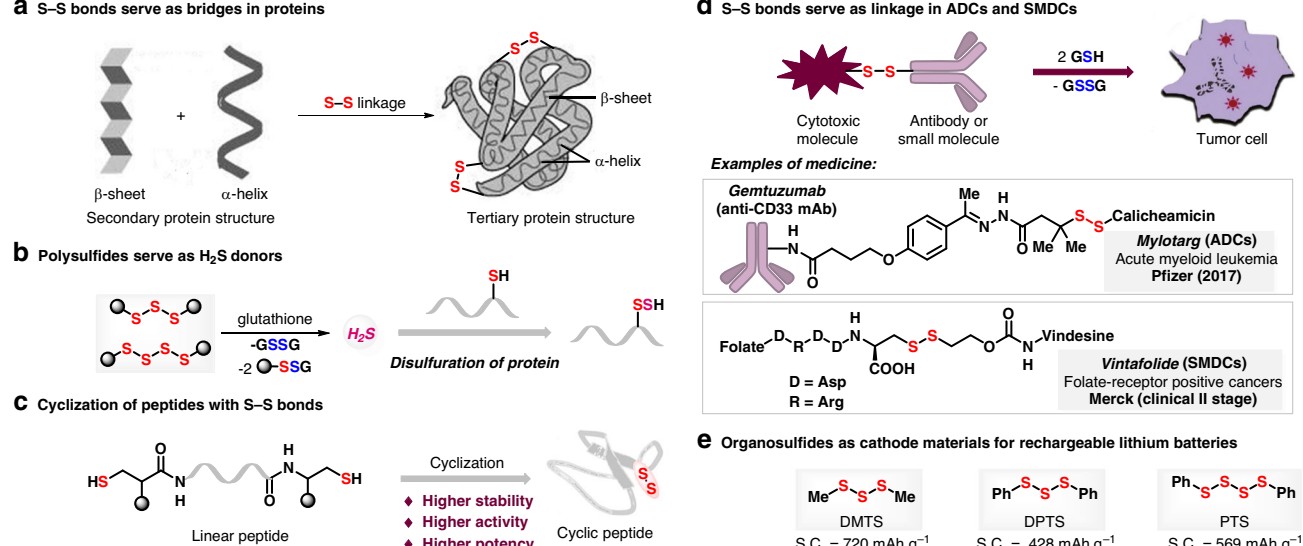

**Fig. 1 The sulfur-sulfur bonds in biological, pharmaceutical, and material fields. a** S–S bonds serve as bridges in proteins. **b** Polysulfides serve as $H_2S$ donors. **c** Cyclization of peptides with S–S bonds. **d** S–S bonds serve as linkage in ADCs and SMDCs. **e** Organosulfides as cathode materials for rechargeable lithium batteries.

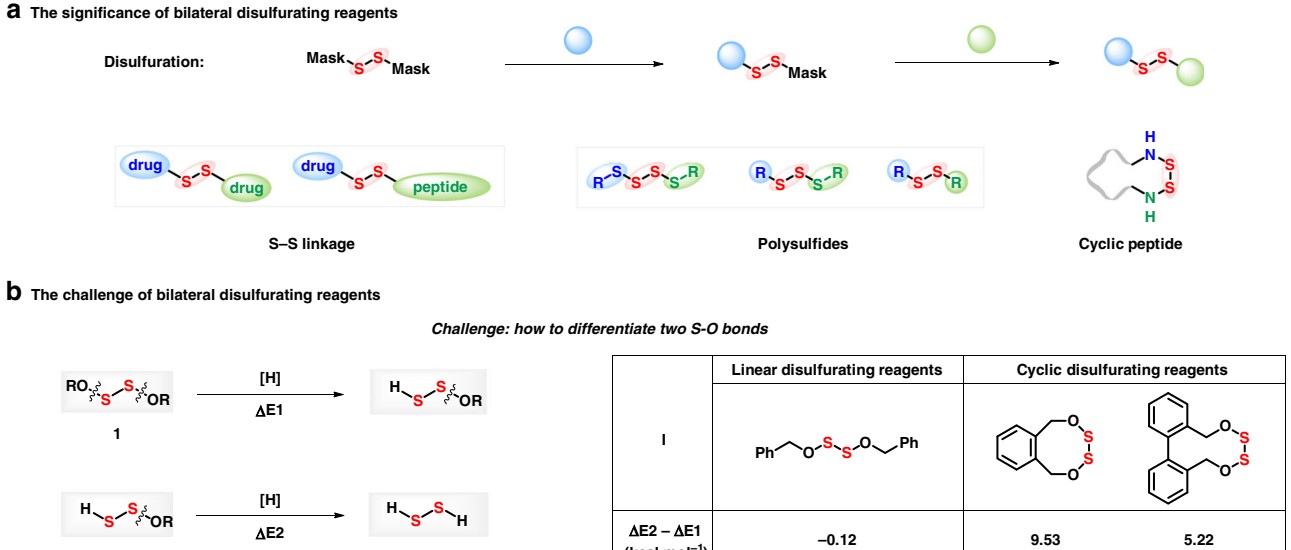

**Fig. 2 The significance and challenge of bilateral disulfurating reagents. a** The significance of bilateral disulfurating reagents. **b** The challenge of bilateral disulfurating reagents.

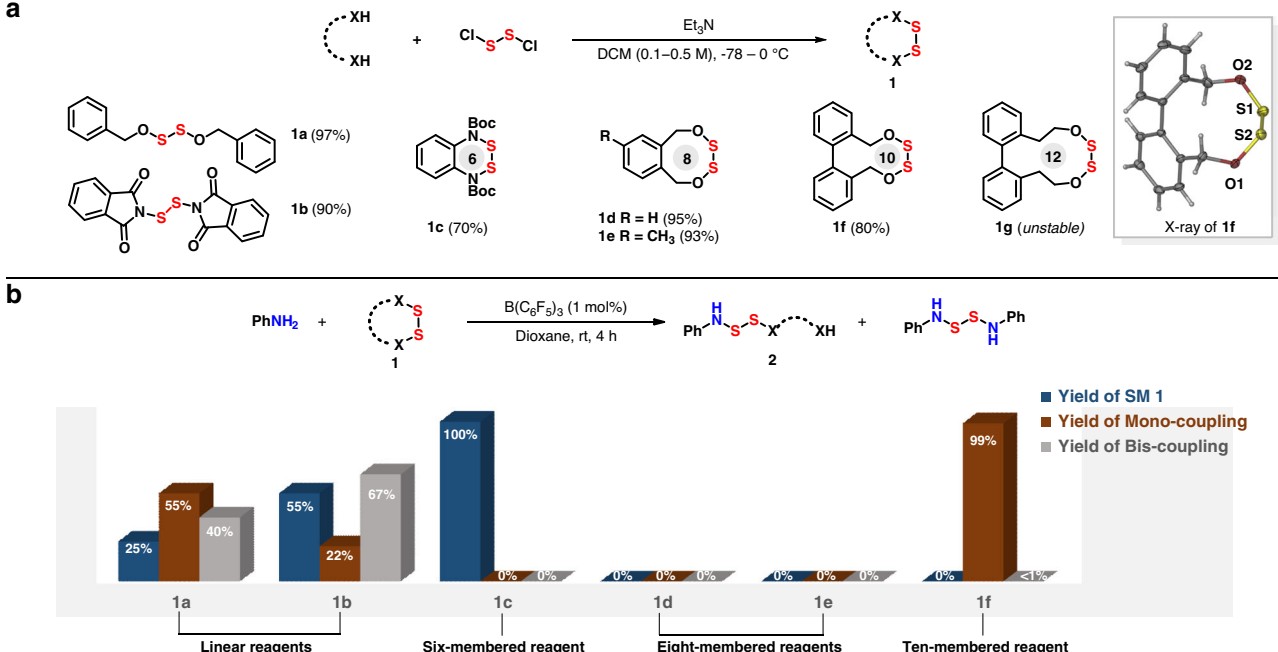

**Fig. 3 Screening of bilateral disulfurating reagents. a** Conditions of disulfurating reagents synthesis. Yields are based on a 10 or 20 mmol scale reaction after silica gel flash chromatography. **b** Sequent reactivity of disulfurating reagents. Aniline (0.10 mmol), 1 (0.105 mmol), B(C$_6$F$_5$)$_3$ (0.001 mmol, 1 mol%), 1,4-dioxane. Isolated yields.

**3o**). Notably, sulfamethazine and sulfamethoxazole could be successfully linked with different peptides in good yields (**3p** and **3q**), which displayed a great potential for the synthesis of SMDCs drugs. Besides, the antibacterial sulfamethazine and cinacalcet, a kind of calcimimetics, could be connected efficiently in good yield (**3r**).

With this strategy, amines could be cross-linked with mercaptans via disulfur motif, affording aza-trisulfides **4** smoothly. Both electron donor and acceptor substituted anilines were applied in the connection compatibly (**4a–4g**). Weak nucleophilic thiophenol, straight-chain dodecamercaptan, electron-rich furfuryl mercaptan, electron-deficient 2-mercaptopyrimidine, and even cysteine could

be successfully introduced in this connection to afford aza-trisulfides (**4h–4m**). Tryptamine, peptide, amines with sensitive enamine structure, even sulfonamides like sulfamethazine, sulfacetamide and sulfamethoxazole were cross-linked with thiols by disulfur perfectly (**4n–4s**), which supplies an efficient protocol for drug-linkage. Interestingly, we successfully synthesized an aza-trisulfide with a long chain of thirty-four-atoms via this method (**4t**). Tripeptides like H-Ala-Phe-Lys-OMe could be cyclized to form 15-membered cyclic peptides **5a** and tetrapeptides like H-Ala-Phe-Trp-Lys-OMe could be cyclized to form 18-membered cyclic peptides **5b** under the standard conditions (Fig. 5).

**Fig. 4 Coupling with amines.** Amine (0.10 mmol), **1f** (0.105 mmol, 1.05 equiv), B(C$_6$F$_5$)$_3$ (0.001 mmol, 1 mol%), 1,4-dioxane, 4 h, then aliphatic amine or mercaptan (0.11 mmol, 1.1 equiv), Li$_2$CO$_3$ (0.1 mmol, 1.0 equiv). Isolated yields. *Ad* Adamantyl. [a]MeCN instead of dioxane.

**Syntheses of aza-disulfides 6, trisulfides 7 and disulfides 8.** Furthermore, we established the cross-linkage between carbon and nucleophiles with phenyl boric acid and bilateral disulfurating reagent **1f** as coupling partner first (Fig. 6). With the optimized conditions, mono-coupling was obtained in 84% yield (for details see the Supplementary Table 2). Investigating on nucleophiles, diverse aromatic rings were cross-linked with amines, mercaptans, and electron-rich aromatics modularly,

affording aza-disulfides, trisulfides, and diaryl disulfides, respectively. The arylboronic acids substituted with electron-withdrawing and -donating functional groups afforded the corresponding aza-disulfides readily (**6a–6f**, **7b–7g**, and **8h**). Arylboronic acids derived from L-tyrosine and estrone were compatible in the cross-linkage, affording a pathway to late-stage modification of natural products (**6g** and **7h**), though the slow rate of transmetallation of boric acid with Cu[III] brought about

**Fig. 5 Coupling with linear peptides.** Diamine (0.2 mmol), **1f** (0.2 mmol), DCM, B(C₆F₅)₃ (0.01 mmol, 5 mol%), 8 h. Isolated yields. Ring sizes are listed with a gray background.

**Fig. 6 Mono-coupling with arylboronic acids.** Conditions: Arylboronic acid (0.15 mmol, 1.5 equiv), **1f** (0.1 mmol), Cu(MeCN)₄PF₆ (0.01 mmol, 10 mol%), 2,2′-bpy (0.02 mmol, 20 mol%), DCM (1 mL), then NuH (0.12 mmol, 1.2 equiv), B(C₆F₅)₃ (0.001 mmol, 1 mol%). Isolated yields. [a]PhMe (1 mL) as solvent in second step.

insufficient efficiency in the first step. The scope of amine is quite broad when it is served as a nucleophile. Anilines (**6a** and **6b**), aliphatic amines (**6c–6e**), amino-acid esters (**6f** and **6g**), and antibiotic sulfamethazine (**6h**) were all efficiently transformed to the corresponding aza-disulfides in moderate yields. Trisulfides could be easily obtained when mercaptans were applied in the cross-coupling. Arylthiophenol like 2-mercaptopyrimidine provided diaryl trisulfide (**7a**). Other thiols even containing hydroxyl (**7b**) and triethoxysilyl ether (**7h**) could afforded trisulfides in moderate yields. Sterically bulky aliphatic thiols, such as *tert*-butylthiol and 1-adamantanethiol, showed great reactivity in this reaction (**7e** and **7f**). Furthermore, cysteine derivatives were successfully converted to trisulfide derivatives (**7d**). Diaryl disulfides were generated when electron-rich aromatics were accommodated under the standard conditions. (+)-δ-Tocopherol, a kind of vitamin E, could be disulfurated directly despite of the presence of free hydroxyl group (**8b**) under nitrogen atmosphere. Indole derivatives were excellent reactants even there is a free amino group (**8c**). Heterocycles like thiophene could be connected in the reaction as well (**8g**). The 2-position of *N*-methyl pyrrole possesses sufficient reactivity in the reaction (**8h**). The structure of **8f** was further confirmed through X-ray analysis.

**Synthesis of tetrasulfides 9**. Unsymmetrical tetrasulfides was a challenging subject in organic synthesis, but the connection between two different mercaptans with **1d** as a disulfurating reagent afforded the desired tetrasulfides highly efficiently, owing to large difference betwen two S-O bonds of eight-membered **1d** (9.53 kcal mol$^{-1}$) (for details see the Supplementary Table 3). The unsymmetrical tetrasulfide linkage was comprehensively investigated in Fig. 7. Pyrimidine and pyrazine can be easily accomodated under the standard conditions (**9a–9c**). The structure of **9a** was further confirmed through X-ray analysis as a linear tetrasulfide. Penicillamine and cysteine, two different amino acids, were cross-linked with tetrasulfur fragment via this method (**9d**). Cysteine (**9e**), tripeptide (**9f**), and even glucosinolate (**9g**) could be cross-linked with 1-adamantanethiol, forming unsymmetrical tetrasulfides. Sensitive thiols, which even contained hydroxyl and triethoxysilyl ether, could afforded tetrasulfides (**9h**).

Remarkably, volatile and low-polar allicin analog was modularly provided when propanethiol and allyl mercaptan were used as nucleophiles (**9i**).

Bilateral reagents **1d** and **1f** are odorless and stable solid stored at −10 °C regardless of air and water. No decomposition was observed even after 5 months, whereas they will deteriorate at room temperature after 24 h. With these designed bilateral reagents, we have established six different kinds of polysulfides, most of which are quite stable under room temperature except aza-trisulfides. They need to be stored in fridge (−10 °C) for long-term preservation. Diaza-disulfides, aza-trisulfide, aza-disulfide, and tetrasulfides are fragile to acidic conditions.

S$_2$Cl$_2$, a common disulfur structure, hardly achieves multiple heteroatom hybrid connection on account of its fractious activity and strong acidity. Taking synthesis of **9a** as an example, the selectivity of **9a** to **9a-S$_5$** is 2.5:1 when S$_2$Cl$_2$ was involved, much lower than 15:1 afforded by our reagent **1d**. Besides, there is a huge gap between the efficiency afforded by S$_2$Cl$_2$ and **1d** (8% vs 70%) (Fig. 8a). Di(1-phthalimidyl)disulfane (**1b**) reagent developed by Harpp[30], which avoids the disadvantage of acidity with S$_2$Cl$_2$, still remains less selective and efficient owing to the nondistinctive S-N bonds. For instance, Harpp's reagent gave a mono-coupling product only in 30% and a bis-coupling byproduct in 60% in the first step coupling with aniline. Optimistically, quantitative yield of **2f** could be obtained with our reagent **1f** (Fig. 8b).

## Discussion

In summary, based on S-O bond dissociation energy nuance, a series of mesocyclic bilateral disulfurating reagents were designed for constructing six species of unsymmetrical polysulfides. Disulfides, trisulfides and tetrasulfides can be accurately achieved with amines, mercaptans, arylboronic acids, and electron-rich aromatic molecules. A considerable range of significant life-related molecules, such as sulfonamides, amino acids, peptides, glucosinolate, vitamin E, and estrone could be cross-linked at will with disulfur bridge to form varieties of diverse functional molecules, which showcases great potential for application of SMDCs and ADCs. Readily available linear

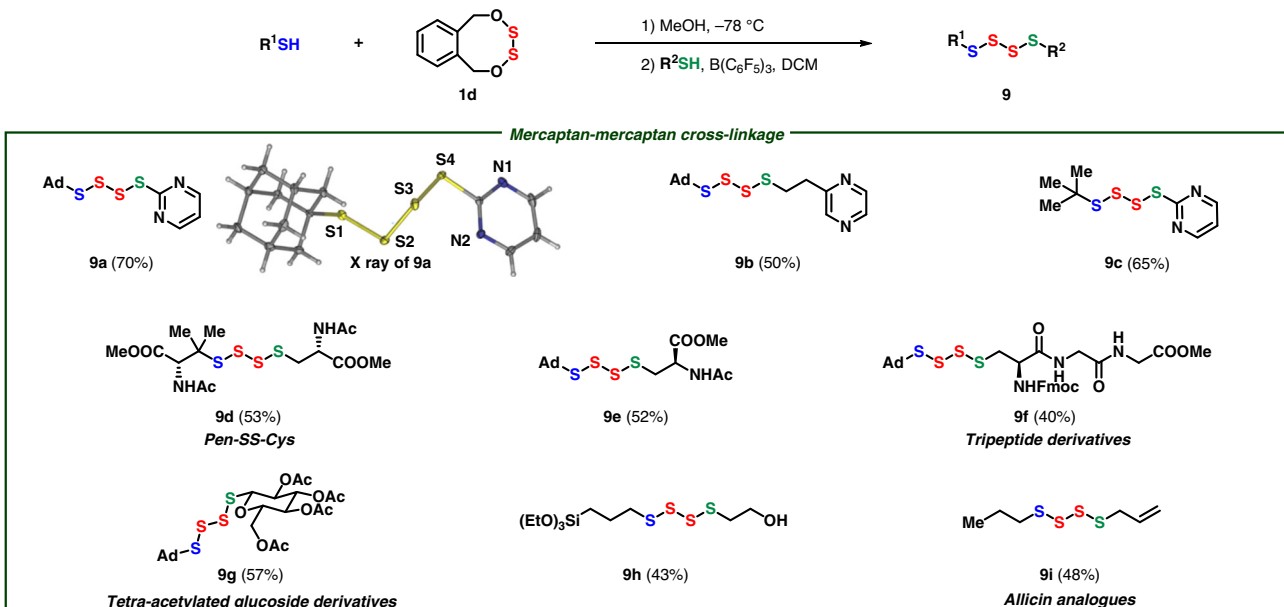

**Fig. 7 Mono-coupling with mercaptans.** Conditions: Thiol (0.1 mmol), **1d** (0.12 mmol, 1.2 equiv), MeOH (2 mL), −78 °C, then R$^2$SH (0.11 mmol, 1.1 equiv), B(C$_6$F$_5$)$_3$ (0.001 mmol, 1 mol%), DCM (1 mL). Isolated yields.

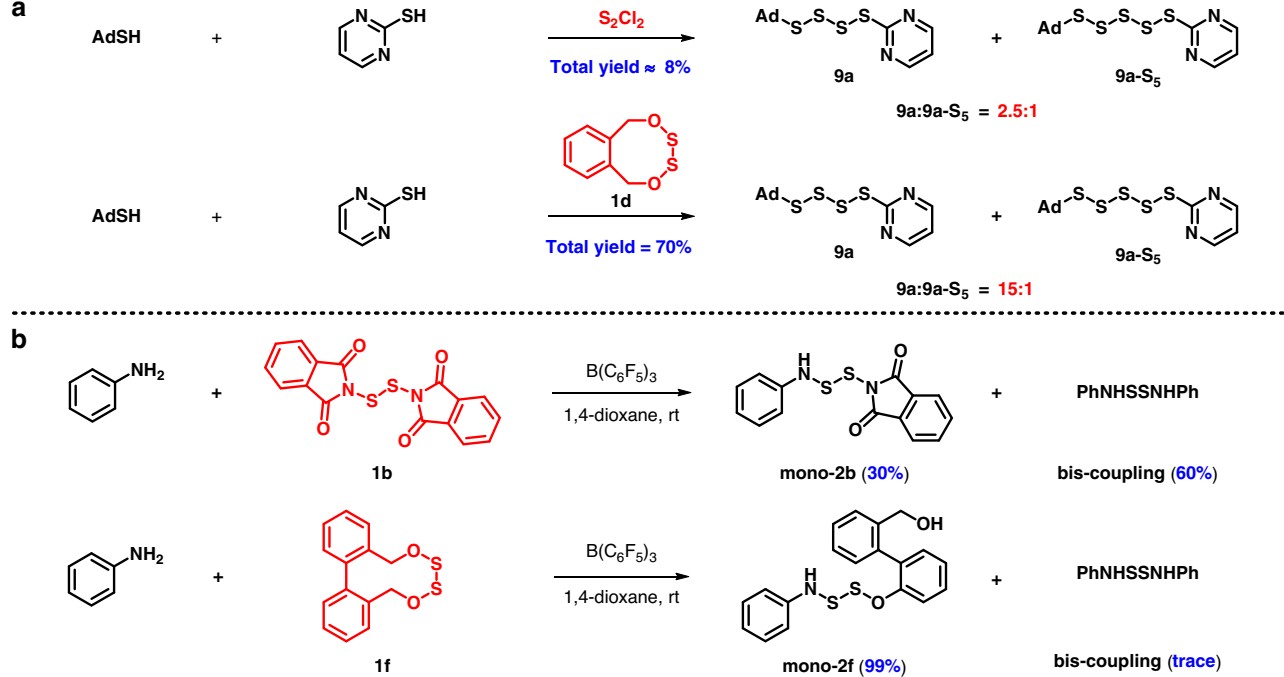

**Fig. 8 Comparison with previous reagents. a** Comparison with $S_2Cl_2$. **b** Comparison with Harpp's reagent **1b**.

peptide precursors can be tied up with disulfur fragment to form unique cyclic peptides with a tetraheteroatomic motif. Drug discovery with polysulfides is undergoing in our laboratory.

## Methods

**General procedure for syntheses of diaza-disulfides 3 and aza-trisulfides 4.** To a Schlenk tube were added amine (0.1 mmol, 1.0 equiv), $B(C_6F_5)_3$ (0.5 mg, 1 mol %),**1f** (29.0 mg, 0.105 mmol), and 1,4-dioxane (0.25 mL), the mixture was stirred at r.t. for 4 h to obtain **2**. After amine was consumed, another amine (0.12 mmol, 1.2 equiv) or thiol (0.11 mmol, 1.1 equiv) and $Li_2CO_3$ (7.4 mg, 0.1 mmol) were added to the mixture. The mixture was stirred at r.t. for 8–24 h before it was concentrated under vacuum. Purification by column chromatography afforded the desired product **3** or **4**.

**General procedure for syntheses of aza-disulfides 6, trisulfides 7, and disulfides 8.** To a Schlenk tube were added arylboronic acid (0.15 mmol, 1.5 equiv), **1f** (29 mg, 0.10 mmol), $Cu(MeCN)_4PF_6$ (3.7 mg, 10 mol%), 2,2′-bpy (3.1 mg, 20 mol %), and redistilled $CH_2Cl_2$ (1 mL), the mixture was stirred at r.t. for 10 h under $N_2$ atmosphere. After **1f** was consumed, another nucleophile (0.12 mmol, 1.2 equiv) was added to the mixture. The mixture was stirred at r.t. for 8 h under air before it was concentrated under vacuum. Purification by column chromatography afforded the desired product **6**, **7**, or **8**.

**General procedure for syntheses of tetrasulfides 9.** To a solution of **1d** (24.0 mg, 0.12 mmol) in MeOH (1 mL) was added thiol (0.1 mmol, 1.0 equiv) in MeOH (1 mL) dropwise at −78 °C, then the mixture was stirred at −78 °C for 30 min before MeOH was removed under vacuum. $CH_2Cl_2$ (1 mL), another thiol (0.11 mmol, 1.1 equiv) and $B(C_6F_5)_3$ (0.5 mg, 0.001 mmol, 1 mol%) was added to the mixture at r.t. for 4 h under air before it was concentrated under vacuum. Purification by column chromatography afforded the desired product **9**.

## Data availability

The X-ray crystallographic coordinates for the structures reported in this study have been deposited at the Cambridge Crystallographic Data Centre (CCDC), under deposition number CCDC-1941481 (**1f**), 1941479 (**3a**), 1941480 (**4d**), 1941478 (**8f**), and 1941482 (**9a**). These data can be obtained free of charge from the Cambridge Crystallographic Data Centre via www.ccdc.cam.ac.uk/data_request/cif. Source data are provided with this paper.

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

## Acknowledgements

We are grateful for financial support provided by the National Key Research and Development Program of China (2017YFD0200500), NSFC (21971065, 21722202, 21672069), the S&TCSM of Shanghai (18JC1415600), Professor of Special Appointment (Eastern Scholar) at Shanghai Institutions of Higher Learning, National Program for Support of Top-notch Young Professionals, and Innovative Research Team of High-Level Local Universities in Shanghai.

## Author contributions

X.J. conceived the idea and supervised the whole project. J.X. designed and carried out the experiments. X.J. and J.X. discussed the results, contributed to writing the manuscript, and commented on the manuscript. All authors approved the final version of the manuscript for submission.

## Competing interests

The authors declare no competing interests.
