## [Peer Review File · Nature Communications]

REVIEWER COMMENTS

Reviewer #1 (Remarks to the Author):

Jiang and coworker described an interesting bilateral disulfurating reagent for diverse polysulfides synthesis bearing various C-, N-, or S-nucleophiles. The strategy could be carried out with one-pot operation under mild conditions. The new design of mesocyclic bilateral disulfurating reagents is remarkable, and functional molecules are modularly installed on both sides of S-S bond, making a comprehensive assembly. Abundant examples showed that the protocol is practical and compatible of functional groups. So I recommend an acceptance of this work published on Nature Communications, but with revise pointed out below.

1. The trisulfides, tetrasulfides, and SSSNs are originally reported for polysulfide synthesis. The stability of these new compounds should be described in the manuscript.
2. The configuration of chiral products should be checked carefully. For example, the configuration of amino acids is S, while one chiral center in product 4p is R-isomer, which is a mistake in my opinion.
3. In Fig 4, the reaction temperature should be included in the equation.
4. Line 161, Ref 7, the authors and the title are not matched with the journal name, the author needs to check it.
5. In Supplementary Information S25, S28, S31, the PhHB(OH)₂ and ArHB(OH)₂ in reaction equations need to be corrected to PhB(OH)₂ and ArB(OH)₂.
6. The references on the synthesis of unsymmetrical polysulfides should be cited in the manuscript.

Reviewer #2 (Remarks to the Author):

This manuscript by Jiang and coworkers revealed a kind of mesocyclic bilateral disulfurating reagents was designed for constructing unsymmetrical polysulfides, including disulfides, trisulfides and even tetrasulfides, contributing to a big progress in polysulfuration. The reagents designed by authors enabled them to be rather useful and flexible as a universal disulfur linker for cross-linking among different types of nucleophiles, including C, N and S. In each type of nucleophile, a broad substrate scope was observed and many impressive polysulfides, such as macrocyclic peptides were successfully synthesized using this method. The bilateral disulfidation introduced in the manuscript could bring great convenience not only in organic chemistry, but also in chemical biology and drug discovery. In addition, the experiments conducted and characterization data of compounds in this manuscript were convincing.

In conclusion, I think this work is novel and significant, and matches the high criteria of Nature Communications. Accordingly, I propose accepting it after minor modifications.

1) It is better to cite some latest references on developing sulfur reagents in the background.

2) "Ad" should be explained at its first appearance, please add.

3) Some typos should be revised. In Page 4, "2f" should be "2e". In Page 6, there is no corresponding structure in Figure 4 which is pertinent to the statement "Sterically bulky 1-adamantanethiol", please remove this statement. In Page 7, the last line, "triethylsilyl ether (7h)" should be "triethoxysilyl ether (7h)", a similar issue existed in Page 9 should also be resolved. In Page 7, "...when propanethiol and allyl mercaptan were used as electrophiles (9i)", "electrophiles" should be "nucleophiles".

4) In Fig. 4, the format of "tBu" in compound 3i, 3j and 3k needs to be unified. In Supplementary Information, the description in parentheses needs to be unified (mole, equivalent, and mass) for the general procedure of compounds.

Reviewer #3 (Remarks to the Author):

INTRODUCTION TO THE PROBLEM BEING ADDRESSED

This interesting submission by Professor Jiang's group, as a follow-up to their 2018 Nature Communications paper, reports on some interesting advances in the area of the synthetic organic chemistry of polysulfide construction. The field has been actively pursued for some decades (referred to hereon as the "first phase"), notably through the endeavours of Professor David Harpp of McGill University in Canada but has received a resurgence in the last five years or so primarily due to the importance of the polysulfide linkage in bioactive materials. Certainly, Professor Jiang has been one of the pioneers responsible for the resurgence by virtue of his new thinking on an old problem, which of course is quintessentially the way science research progresses. Backtracking a little, the main challenges that emerged from the "first phase" of research when targeting the various types of polysulfides (notably di, tri and tetra), can be summarised by the following three issues:

- 1) The homo/hetero coupling problem of sulfonyl methodology (as the most prevalent methodology available) when converting R₁SH to R₁SSR₂, concerning the challenge of the reaction product being contaminated with homodimers R₁SSR₂ and R₂SSR₂.
- 2) The instability and purification of starting reagents and reaction intermediates. For instance, S₂Cl₂ is a popular reagent (directly or indirectly) available for synthesising trisulfides and tetrasulfides, but is difficult to obtain in a pure form that is uncontaminated with SCl₂. Similarly, R(S)_nL_g sulfonylating agents have a range of stabilities, a feature that is often not adequately and objectively revealed in publications.
- 3) Following on from point 2), and specifically in context to the disulfanyl type in this paper (which is of the type L_gSSL_g where L_g = a leaving group), little is known about mechanism. In particular, it is known that RSSL_g species (eg with -SL_g = -thiosulfonates - see Harpp 1979 JOC paper uploaded - and halides; imides to a lesser extent) are unstable and tend to extrude sulfur, resulting in mixtures. This is a particular problem when the R groups in question are non-polar, in view of chromatography separation difficulties.

SPECIFIC APPRAISAL OF THIS SUBMISSION

So let's now go through the three points above and see how Professor Jiang gets on:

Point 1

Definitely an advance on the homo/hetero coupling problem has been made through use of a cyclic 10-membered bis-alkoxy disulfanyl template (1f), in which the first substitution is more energetically favourable (exothermic) than the second substitution due to ring-strain release. His choice of a cyclic dialkoxy-disulfanyl template is a clever one, because there have been very few examples of using an OR group as a leaving group within sulfonylation methodology; plenty for S- and N-based leaving groups but not ones based on O. Moreover, Professor Jiang has cleverly identified a Lewis acid activator in the form of a triarylborane, which in conjunction with his earlier finding that activated carbon-based nucleophiles (with Lewis acid activation) and non-activated ones (under cross-coupling conditions with a copper (I) catalyst and a boronic acid) can be used as nucleophiles in the sulfonylation process, means that all three nucleophile types of N, C and S can be coupled into the polysulfide structure.

However, that said, although the selectivity is amply demonstrated in a model study, it is not quantified in a real reaction. Groups are chosen so as not to run into chromatographic separation difficulties so that the product (as verified by the SI data) can be isolated in pure form. If, for instance, we take one of the early products, let's say 3d (Figure 4) from the bis-sulfenamides, in which there are p-vinylanilino and 3,4-dimethoxybenzyl groups, this surely wouldn't have been a problem to separate from homodimers. The model study demonstrates a 99% yield, which appears to solve the homodimer issue (since only one equivalent or slightly less of the nucleophile is

added). So why is the yield for 3d only 55%? Is it because step 1 is now no longer as selective or is step 2 the problem? In some cases, the yield drops further (see bis-sulfenamides 3l-3n with 35%-42% yields). Like I said, this doesn't present a problem in the chromatography, because the groups are chosen with enough polarity, but it would be if the groups being added through substitution were less polar. For instance, turning to the tetrasulfide analogues derived from using two sequential thiols, the yields for 9h and 9i are only 43% and 48% respectively. 9h has a polar handle (primary hydroxyl group), so chromatographic purification wouldn't have been a problem, but the allicin analogue 9i is non-polar and there may well have been some challenges, which certainly wouldn't be easily overcome on scale-up. With a yield of 48% one can't say that there is any real improvement on using a double substitution of the standard S₂Cl₂. Overall, the range of examples presented haven't proved in real (quantified) terms (see point 3 about sulfur extrusion) that the homo/heterodimer problem of point 1 has been overcome. It looks that way from the model, but then why are yields so low in many cases?

Point 2

The key template 1f is prepared in high yield (80%) by reacting a diol with S₂Cl₂ as the sulfur source (a huge drawback, I must say!!), and is isolated through a conventional work-up followed by column chromatography. It is known that RSSOMe species are quite labile (see the Heimgartner paper attached), yet nothing specific is mentioned in the text as to the shelf-life and handlability of 1f (as a cyclic R'OSSOR species). It appears to be more stable certainly, but why should it be more stable than 1g, for instance? What science rationale can be presented on this aspect? Could this also be due to sulfur extrusion – see point 3? On an industrial or Fine Chemicals scale, could 1f be prepared and handled?

Point 3

By and large this links to point 2, but does introduce the important issue of mechanism. Indeed, this point 3 highlights the most damning criticism of Professor Jiang's submission. If I had to hazard a guess and put forward an explanation as to why several of the yields are low to moderate across several of the combinations, I would opt for sulfur extrusion. In the second and more difficult substitution step requiring room temperature (also step 1, but sometimes this can be carried out at lower temperature), one has to question what the mechanism is. The OR group is complexed to the boron - does this then not result in a transient sulfenylium ion reaction intermediate as NuSS⁺ (Ar)₃B-OR⁻, kind of SN1 style? If so, then sulfur extrusion is likely, due to a superior resonance stabilisation offered by Nu, particularly if Nu is nitrogen or sulfur: (NuSS⁺ → Nu+=S + S; thereafter, the second nucleophile adds). An SN2-type mechanism would also likely follow the same route. Such a sulfur extrusion would make for a difficult chromatographic separation of compounds differing by virtue of only a single sulfur atom. The need for room temperature (increasing (negatively) the free-energy entropy factor of -TΔS, making ΔG more negative overall) encourages the elimination, meaning that a lack of reactivity, though beneficial in terms of stability, leads to the reaction downfall because low reaction temperatures that would limit S extrusion can't be used. The failure to account for many of the low yields, as well as offering no mechanistic insight, significantly reduces the value of this submission. What is needed is to take some of the low-yielding reactions and objectively report on the reality of the product spectrum, ie to determine the structures of all products produced. A combination of non-polar groups here would be the acid test as it always has been, for reasons already given about separation difficulties. This approach would generate data that would quickly help to resolve some of the attendant mechanistic issues, as well as generate a much-needed mechanistic profile.

OVERALL CONCLUSION

Overall, while the templated idea appears to be a possible step forward in this field, there are too many unanswered questions that need to be addressed. More emphasis needs to be placed on generating quality data on smaller libraries, as this sweeping panoramic presentation lacks underlying mechanistic quality, which sends a very mixed message for this prestigious journal. I therefore recommend a rejection and a rethink. NB:(¹³C NMR data needs to be given to one decimal place only).

The detailed replies to reviewer's question are listed as below:

Reviewer #1: "recommend its publication"

Q1: The trisulfides, tetrasulfides, and SSSNs are originally reported for polysulfide synthesis. The stability of these new compounds should be described in the manuscript.

A1: Thank you for your suggestion. The description of polysulfide stability was added in Page 9 as follow.

"Bilateral reagents **1d** and **1f** are odorless and stable solid stored at -10 °C regardless of air and water. No decomposition was observed even after 5 months, while they will deteriorate at room temperature after 24 h. With these designed bilateral reagents, we have established six different kinds of polysulfides, most of which are quite stable under room temperature except aza-trisulfides. They need to be stored in fridge (-10 °C) for long-term preservation. Diaza-disulfides, aza-trisulfide, aza-disulfide and tetrasulfides are fragile to acidic conditions."

Q2: The configuration of chiral products should be checked carefully. For example, the configuration of amino acids is S, while one chiral center in product **4p** is R-isomer, which is a mistake in my opinion.

A2: Thanks for the careful revision. We have rechecked all the configurations of chiral products. The configuration of lysine in product **4p** has been corrected to S-configuration in Fig. 4 and Page S20, S128, S129 of Supplementary Information. Tryptophan has been corrected from racemate to S-configuration in product **6g** in Fig. 6 and Page S28, S157, S158 of Supplementary Information.

Q3: In Fig 4, the reaction temperature should be included in the equation.

A3: The reaction is at room temperature, which has been added in the equation of Fig. 4.

Q4: Line 161, Ref 7, the authors and the title are not matched with the journal name, the author needs to check it.

A4: We have rechecked all the reference and made correction of ref. 7 as follow:

“Staben, L. R. et al. Targeted drug delivery through the traceless release of tertiary and heteroaryl amines from antibody-drug conjugates. *Nat. Chem.* **8**, 1112–1119 (2016).”

Q5: In Supplementary Information S25, S28, S31, the $PhHB(OH)_2$ and $ArHB(OH)_2$ in reaction equations need to be corrected to $PhB(OH)_2$ and $ArB(OH)_2$.

A5: Thanks for the careful revision. The typos on Pages S25, S28, and S31 of Supplementary Information have been corrected to $PhB(OH)_2$ and $ArB(OH)_2$.

Q6: The references on the synthesis of unsymmetrical polysulfides should be cited in the manuscript.

A6: References 21-24 have been added according to the Reviewer’s suggestion:

21. Park, C. M. et al. 9-Fluorenylmethyl (Fm) disulfides: biomimetic precursors for persulfides. *Org. Lett.* **18**, 904–907 (2016).

22. Wang, W. et al. Cu-catalyzed electrophilic disulfur transfer: synthesis of unsymmetrical disulfides. *Org. Lett.* **20**, 3829–3832 (2018).

23. Zou, J. et al. Phthalimide-carried disulfur transfer to synthesize unsymmetrical disulfanes via copper catalysis. *ACS Catal.* **9**, 11426–11430 (2019).

24. Gao, W., Tian, J., Shang, Y. & Jiang, X. Steric and stereoscopic disulfide construction for cross-linkage *via N*-dithiophthalimides. *Chem. Sci.* **11**, 3903–3908 (2020).

Reviewer #2: “accept it after minor modifications.”

Q1: It is better to cite some latest references on developing sulfur reagents in the background.

A1: Thank you for the reviewer’s suggestion. References 21-24 have been added on developing sulfur reagents in the background as follow.

21. Park, C. M. et al. 9-Fluorenylmethyl (Fm) disulfides: biomimetic precursors for persulfides. *Org. Lett.* **18**, 904–907 (2016).

22. Wang, W. et al. Cu-catalyzed electrophilic disulfur transfer: synthesis of unsymmetrical disulfides. *Org. Lett.* **20**, 3829–3832 (2018).

23. Zou, J. et al. Phthalimide-carried disulfur transfer to synthesize unsymmetrical disulfanes via copper catalysis. *ACS Catal.* **9**, 11426–11430 (2019).

24. Gao, W., Tian, J., Shang, Y. & Jiang, X. Steric and stereoscopic disulfide construction for cross-linkage *via N*-dithiophthalimides. *Chem. Sci.* **11**, 3903–3908 (2020).

Q2: “Ad” should be explained at its first appearance, please add.

A2: “Ad = Adamantyl” have been added to explain the “Ad” in the footnote of Fig. 4.

Q3: Some typos should be revised. In Page 4, “**2f**” should be “**2e**”. In Page 6, there is no corresponding structure in Figure 4 which is pertinent to the statement “Sterically bulky 1-adamantanethiol”, please remove this statement. In Page 7, the last line, “triethylsilyl ether (**7h**)” should be “triethoxysilyl ether (**7h**)”, a similar issue existed in Page 9 should also be resolved. In Page 7, “...when propanethiol and allyl

mercaptan were used as electrophiles (**9i**)”, “electrophiles” should be “nucleophiles”.

A3: Thanks for the reviewer’s careful revision. We have checked the entire manuscript and revised the typos as follow.

1) We have corrected “...subjected to aza-disulfide **2e** under the assistance of weak base...” to “...subjected to aza-disulfide **2f** under the assistance of weak base...” since reagent **1f** is optimized disulfurating reagent in this reaction.

2) We have removed the statement of “Sterically bulky 1-adamantanethiol” in the “Sterically bulky 1-adamantanethiol, weak nucleophilic thiophenol...”.

3) We have corrected “triethylsilyl ether” to “triethoxysilyl ether” in Page 7 and 9.

4) We have corrected “...when propanethiol and allyl mercaptan were used as electrophiles (**9i**)” to “...when propanethiol and allyl mercaptan were used as nucleophiles (**9i**)”.

Q4: In Fig. 4, the format of “tBu” in compound **3i**, **3j** and **3k** needs to be unified. In Supplementary Information, the description in parentheses needs to be unified (mole, equivalent, and mass) for the general procedure of compounds.

A4: Thank you for your suggestion. We have unified the format of “tBu” of compounds **3i**, **3j** and **3k** in Fig.4. And the description in parentheses has been unified to (mass, mole, and equivalent) in Supplementary Information.

Reviewer #3:

Q1: If, for instance, we take one of the early products, let’s say **3d** (Figure 4) from the bis-sulfenamides, in which there are p-vinylnilino and 3,4-dimethoxybenzyl groups,

this surely wouldn't have been a problem to separate from homodimers. The model study demonstrates a 99% yield, which appears to solve the homodimer issue (since only one equivalent or slightly less of the nucleophile is added). So why is the yield for **3d** only 55%? Is it because step 1 is now no longer as selective or is step 2 the problem?

A1: Thank you for your inquiry. The intermediate [N-S-S-O] structure **M1** in the first step has been isolated with a yield of 62%, which was subjected to the second step affording 89% of desired products. It demonstrated that the second step is highly efficient and selective. Furthermore, step 1 was monitored via LC-MS of crude reaction solution in Fig. R1, which exhibited that the selectivity of mono-substituted product **M1** and symmetrical disulfide byproduct is over 20:1. Meanwhile, the high polar interval of LC spectrum (0.2-1.8 min) displayed a broad mixture with m/z between 400 to 800, which was oligomers caused by the vinyl group lowering the yield of step 1. The polymerization can be accelerated via $B(C_6F_5)_3$. (Reference about polymerization accelerated by $B(C_6F_5)_3$: Yang, X., Stern, C. L. & Marks, T. J. Cation-like homogeneous olefin polymerization catalysts based upon zirconocene alkyls and tris(pentafluorophenyl)borane. *J. Am. Chem. Soc.* **113**, 3623–3625 (1991).)

Fig. R1: LC-MS of crude reaction solution of M1.

Q2: In some cases, the yield drops further (see bis-sulfenamides **3l-3n** with 35%-42% yields). ...It looks that way from the model, but then why are yields so low in many cases?

A2: As you mentioned, methyl L-tryptophanate motif in compounds **3l**, **3n**, and **3o**, possessing multiple nucleophilic sites and highly reactive indolyl structure, was the

N-nucleophile with relatively low yields in step 1. However, the biological importance of this structure drives us to crosslink it with other amino acid esters via disulfur bridge, which is an ideal precursor for “H₂S” releasing. Meanwhile, in the cases of **3l**, **3m**, and **4t**, octadecanamine showed poor solubility, inducing the transformation rate of step 2 to be slow and inefficient.

Q3: **9h** has a polar handle (primary hydroxyl group), so chromatographic purification wouldn't have been a problem, but allicin analogue **9i** is non-polar and there may well have been some challenges, which certainly wouldn't be easily overcome on scale-up.

...A combination of non-polar groups here would be the acid test as it always has been, for reasons already given about separation difficulties.

A3: Yes, as you mentioned, this is one of the advantages of our reagents. With our bilateral disulfurating reagents, we could overcome the separation challenge for the synthesis of non-polar allicin analogue **9i**. Since of high-polar stepwise intermediate, column chromatography can be applied after step 1 for removing the non-polar impurities. Followed by highly efficient and selective step 2, pure allicin analogues can be readily obtained. Detailed procedure was in Page S38 of Supplementary

Information.

Q4: With a yield of 48% one can't say that there is any real improvement on using a double substitution of the standard S_2Cl_2 .

A4: 1) Reagent S_2Cl_2 is acidic with HCl releasing when meeting trace of water. However, our stable bilateral disulfurating reagents [O-S-S-O] is neutral, compatible with sensitive functional group and broader substrates.

2) Separation challenges is always symbiotic with the application of S_2Cl_2 due to the non-polar molecules. However, our bilateral disulfurating reagents [O-S-S-O], since of high polar hydroxyl-containing [X-S-S-O] intermediate, makes tough cases easy to be isolated.

3) Most of unsymmetrical tetrasulfide syntheses using S_2Cl_2 are low yields and difficult to purify due to the absence of selectivity, which is difficult to apply to high-efficiency demanding drug discovery and fine chemicals. (B. Czepukojc et al. *Phosphorus Sulfur* **2013**, 188, 446; Allah, D. R. et al. *Int. J. Oncol.*, **2015**, 47, 991)

Q5: ...nothing specific is mentioned in the text as to the shelf-life and handlability of **1f**.

A5: Thank you for your suggestion. The description of shelf-life and handlability were added in the Page 9.

“Bilateral reagents **1d** and **1f** are odorless and stable solid stored at $-10\text{ }^\circ\text{C}$ regardless of air and water. No decomposition was observed even after 5 months, while they

will deteriorate at room temperature after 24 h.”

Q6: It (**1f**) appears to be more stable certainly, but why should it be more stable than **1g**, for instance? What science rationale can be presented on this aspect? Could this also be due to sulfur extrusion.

A6: Experimentally, we discovered fast decomposition happened when 12-membered reagent **1g** is in the solution with trace amount of acid. Based on the MM2 calculation in Fig. R2, 12-membered reagent **1g** is much easier to be protonated than 10-membered reagent **1f**. Through the analysis of energy diagram, facile decomposition of reagent **1g** is preferential in thermodynamics, while high activation energy of coordination from acid deterred the decomposition of reagent **1f**.

Fig. R2: Energy difference between protonation of reagent **1f** and **1g**.

Q7: On an industrial or Fine Chemicals scale, could **1f** be prepared and handled?

A7: Considering the reviewer's suggestion, we prepared reagent **1f** with decagram scale with a steady yield of 78%. The procedure has been added in the Page S5 of Supplementary Information.

Furthermore, compound **3j** was synthesized on a 5 mmol scale with yield of 65%, recovering [1,1'-biphenyl]-2,2'-diylmethanol with yield of 88%, which can be reused for reagent preparation. Procedure and detailed information have been added in Page S14 of Supplementary Information.

Q8: If I had to hazard a guess and put forward an explanation as to why several of the yields are low to moderate across several of the combinations, I would opt for sulfur extrusion. In the second and more difficult substitution step requiring room temperature (also step 1, but sometimes this can be carried out at lower temperature), one has to question what the mechanism is. The OR group is complexed to the boron – does this then not result in a transient sulfenium ion reaction intermediate as $\text{NuSS}^+(\text{Ar})_3\text{B-OR}^-$, kind of $\text{S}_{\text{N}}1$ style?

A8: Thank you for your helpful mechanism discussion. $\text{S}_{\text{N}}1$ style is a possible transformation explanation for our reagents. However, control experiments show greater possibility of $\text{S}_{\text{N}}2$ process. First of all, the transformations of step 2 with **6f**, **6g** and **6h** refused to be realized when weaker nucleophilic amines were subjected in

the solvent of DCM. However, when the solvent was altered to nonpolar solvent toluene, the desired substitution happened smoothly. Theoretically, polar solvent, which can stabilize a transient sulfenylium ion, is favorable for S_N1 reaction. On the contrary, nonpolar solvent usually supports S_N2 reaction.

Reference of solvent effect: (Bo, X. A computational method for solvent effects on the (Cl+CH₃Br → CH₃Cl+Br⁻) S_N2 nucleophilic substitution reaction. *Mod. Sci. Instr.* **3**, 89–91 (2010); Smith M. B., March, J. March's advanced organic chemistry, reactions, mechanisms and structure, 6th ed by John Wiley & Sons, Inc.: New Jersey, 2007; Chapter 10; p. 432)

Q9: If so, then sulfur extrusion is likely, due to a superior resonance stabilization offered by Nu, particularly if Nu is nitrogen or sulfur: (NuSS⁺ → Nu⁺=S + S; thereafter, the second nucleophile adds). An S_N2-type mechanism would also likely follow the same route. Such a sulfur extrusion would make for a difficult chromatographic separation of compounds differing by virtue of only a single sulfur atom.

A9: Thank you for your kind proposal for the possibility. However, in-situ LC-MS detection and NMR spectrum never show sulfur extrusion products nor superior resonance stabilized intermediate whether Nu is nitrogen or sulfur.

1) Sulfur extrusion has never been detected during the synthesis of **4j**. But we noticed an interesting phenomenon detected by LC-MS during **4j** synthesis, which lowered the yield of **4j**. (Possible mechanism is shown as below. Final product **4j** is liable to **BP2** via intramolecular cyclization gradually, attacked by **S2** to generate **BP4** or by **S1** to generate **BP3**. **BP3** could generate cyclic polysulfide of **BP6**, attacked by **S1** generating pentasulfide **BP5**.)

XJH-12-133-1 Sm (Mn, 2x3)

3: Diode Array
Range: 2.113
Area

Fig. R3: LC-MS of 4j synthesis.

- 2) Sulfur extrusion has never been detected during the synthesis of 3c in Fig. R4. But we noticed that symmetrical **BP5** (around 10%) was generated via LC-MS of 3c synthesis, due to the slightly excessive of CyNH₂ (1.2 eq.).

Fig. R4: LC-MS of **3c** synthesis.

3) Sulfur extrusion has been never observed during the synthesis of **9a** with reagent **1d** compared with reagent **1c** in Fig. R5, when Nu is sulfur. (From ^1H NMR spectrum of our reagent **1d** transformation, no S extrusion was detected with pure desired product in 70% yield. But S extrusion was found when using reagent **1c**, resulting in the mixture of trisulfide and tetrasulfide.)

Fig. R5: ^1H NMR comparison of **9a** synthesis between reagent **1d** and **1c**.

Q10: The need for room temperature (increasing (negatively) the free-energy entropy

factor of $-T\Delta S$, making ΔG more negative overall) encourages the elimination, meaning that a lack of reactivity, though beneficial in terms of stability, leads to the reaction downfall because low reaction temperatures that would limit S extrusion can't be used.

A10: The reviewer gave us another way to explain S extrusion from thermodynamics. However, S extrusion hardly happened in our reagent transformation under optimized conditions. Furthermore, control experiments don't support the elimination mechanism. In S extrusion, $\Delta S > 0$, $\Delta G = \Delta H - T\Delta S$. ΔG is linear inverse correlation with T, therefore decrease of reaction temperature will limit the S extrusion. In the example of **7d**, all the ingredients of step 2 was detected under $-40\text{ }^{\circ}\text{C}$, $0\text{ }^{\circ}\text{C}$, $25\text{ }^{\circ}\text{C}$ and $40\text{ }^{\circ}\text{C}$ via LC-MS in Fig. R6. With these data, the correlation between selectivity and temperature was exhibited in Fig. R7, which shows there is no linear inverse correlation between reaction temperature and selectivity of trisulfide and disulfide. The entropy increase didn't happen in this reaction, which further proved no S extrusion happen during the transformation at least under optimized conditions.

Fig. R6: LC-MS comparison between trisulfide **7d** and disulfide.

Fig. R7: Fitting curves between selectivity and temperature.

Q11: ^{13}C NMR data needs to be given to one decimal place only.

A11: All ^{13}C NMR data have been revised to one decimal place in Supplementary Information.

REVIEWER COMMENTS

Reviewer #1 (Remarks to the Author):

Overall, the authors have improved the manuscript through addressing the requested referee changes. I think this manuscript is ready for publication in this journal.

Reviewer #3 (Remarks to the Author):

For this stratospheric journal, the revision manuscript is still weak in certain areas, in my opinion, as summarised by the following:

1) It doesn't give any background to the disulfanyl transfer literature based on reagents of the form Lg-S-S-Lg (Lg = leaving group). Notably, it completely ignores Harpp's seminal JACS paper from 1978 (attached) on Lg = phthalimido and others. The same argument applies to any other methods involving Lg-S-S-Lg reagents. Admittedly, Harpp didn't do that much synthetically with his reagents thereafter and certainly didn't achieve anywhere the success achieved with the present work, but that's all the more reason to use this to highlight the present work.

2) Following on from point 1), we need to know the practicalities of this new method so as to be able to compare it with other similar methods belonging to the methodology of double nucleophilic substitution of a Lg-S-S-Lg reagent. In particular, we need to know: i) how efficient the first step is (not just for the model but in general), ii) what the by-products are for this step, and iii) how easy they are to separate. Then, the same appraisal for the second step. If a collective statement is possible covering the two steps, so be it, but the three questions need to be answered in some shape or form (reaction efficiency (both steps), nature of any by-products, ease of purification, including the pivotal RSSOR' intermediate, which is expected to be quite fragile on chromatography and needs to be quantified in some way. The need for answers to these questions is demanded by i) the low to moderate yields in several of the examples, and ii) so as to be able to satisfactorily conclude that this truly is an advance in the field befitting this stratospheric journal. As things stand, it's just a list of results selling the method. The lack of reaction mechanistic appraisal (still missing) was another feature commented on in my first review, and this would emerge naturally if more information on the three questions above were addressed properly.

Reviewer #2:

Q: Overall, the authors have improved the manuscript through addressing the requested referee changes. I think this manuscript is ready for publication in this journal.

A: Thank you for the reviewer's recognition for our revision.

Reviewer #3:

Q1: It doesn't give any background to the disulfanyl transfer literature based on reagents of the form Lg-S-S-Lg (Lg = leaving group).

A1: Thank you for the reviewer's suggestion. Introduction for background of disulfanyl transfer has been reinforced in Page 3 as follow:

“Dialkoxydisulfide²⁸⁻²⁹ and diaminedisulfide³⁰⁻³² have been investigated as sulfur transfer reagents since 1970s. However, unsymmetrical polysulfidation with disulfanyl motif has never been achieved due to the sharp contradiction raised by sequential and selective cleavage of dual S-O(N) bonds.”

28. Tardif, S. L., Williams, C. R. & Harpp, D. N. Diatomic sulfur transfer from stable alkoxy disulfides. *J. Am. Chem. Soc.* **117**, 9067–9068 (1995).
29. Zysman-Colman, E. et al. Crossover point between dialkoxy disulfides (ROSSOR) and thionosulfites ((RO)₂S=S): prediction, synthesis, and structure. *J. Am. Chem. Soc.* **128**, 291–304 (2006).
30. Harpp, D. N., Steliou, K. & Chan, T. H. Synthesis and reactions of some new sulfur transfer reagents. *J. Am. Chem. Soc.* **100**, 1222–1228 (1978).
31. Huang, N., Lakshmikantham, M. V. & Cava, M. P. Thiation reactions of some active carbonyl compounds with sulfur transfer reagents. *J. Org. Chem.* **52**, 169–172 (1987).
32. Graf, T. A. et al. New polymers possessing a disulfide bond in a unique environment. *Macromolecules* **45**, 8193–8200 (2012).

Q2: How efficient is the first step and second step (in the reaction)?

A2: Thank you for your inquiry. According to your constructive suggestion, we have

relaunched the reactions with yields under 55%. The first-step efficiency was detected via ^1H NMR and LC-MS. The second-step efficiency was calculated based on first-step yields and total yields. Experimental procedures and data have been attached in Page S7-S8 and Page S28-S29 of Supplementary Information. Summarized information about efficiency of two steps was shown as follow:

Scheme. A2 Summary of low efficient reactions.

- 1) For **3l**, **3n** and **3o**, isolated yield of the first step is 36%, which is lower than the final yields of **3l**, **3n** and **3o**. It means mono-substituted [NSSO] is not the only intermediate during this transformation, which we addressed the detail in “Question 3”.
- 2) Besides **3l**, **3n** and **3o**, the first steps of **6g** and **7h** are under 50% yields, due to the slow transmetalation of boric acid with Cu[III], which brought about several uncontrollable subsidiary reactions in the system.
- 3) The second step of **3m** gave a yield of 47% owing to the poor solubility of octadecanamine, which resulted in an inefficient transformation.
- 4) The second step of **8b** gave a yield of 44%, since anti-oxidative vitamin E is unstable under air conditions. With nitrogen atmosphere, we have promoted the yield of the second step to 70% and the final yield to 59%.
- 5) The second steps of **8d** and **8e** gave low yields, due to the competitive multiple-position nucleophilicity of indole generating side reactions.

Q3: What the by-products are (in the reaction)?

A3: Thank you for your inquiry. In order to depict the by-products of the reactions resulting in low yields, we monitored full-course of **3l**, first steps of **6g** and **3d**, second steps of **8b** and **8d** via LC-MS. In general, substrates with special functional groups offer extra impact on disulfuration.

1. Substrates with tryptophan motif, such as **3l**, **3n** and **3o**, afforded a low yield of **M1** of 36%, inconsequently lower than the total yields of **3l**, **3n** and **3o**. The LC-MS detection of the first step demonstrated that cyclic disulfide intermediate **M2** was generated due to the nucleophilicity of 2-position of indole. In the second step, cyclic disulfide intermediate **M2** disappeared from LC-MS spectrum. Obviously, intermediate **M2** generating from nucleophilic cyclization of 2-position of indole brought about the unusual experimental results. In the first step, the homocoupling intermediate **M2** yielded symmetrical product **BP1**. In the second step, intermediate **M1** and **M2** afforded product **3l** together, which created the distortion between the yields of two steps. Remarkably, the cross-linkage between biologically and medically important tryptophan and other amino acid is an indispensable exploration for polysulfide peptide drug discovery.

Scheme. A3.1 Pathways for **3l** synthesis.

Fig. A3.1 LC-MS of the first and second steps for 31

2. The yield of the first step of **6g** was 46%, which is much lower than that with other substrate. LC-MS detection shows that the rate of transmetalation of boric acid with Cu[III] was so slow that a large number of subsidiary reactions occurred in the system, such as deprotection (**BP3**), oxidation (**BP4**), protonation (**BP5**), etherification (**BP6**), and secondary coupling (**BP7** and **BP8**).

Fig. A3.2 LC-MS of the first step for 6g

3. The yield of the second step for **8b** was 44%, which is much lower than most of the reaction. LC-MS of this step shows that the peak areas of reactant vitamin E and product **8b** were much smaller than that of the leaving group **BP2**, but there is no other by-product in the LC-MS spectrum. So we suspected that the oxygen in air induced their oxidative decomposition. When we relaunched this reaction under N_2 , the yield of second step of **8b** was increased to 70% and the total yield was increased to 59%. Detailed experimental procedure has been added in Page S36 of Supplementary Information.

Fig. A3.3 LC-MS of the second step for 8b

4. The yields of the second step of **8d** and **8e** were both low as 57%. Therefore, we detected the second step of **8d** by LC-MS and found that owing to the strong nucleophilicity of 2- and 3- position of indole, numbers of subsidiary products were inevitably generated, such as double substituted **BP10** and isomer of **8d**.

Fig. A3.4 LC-MS of the second step for 8d

5. The yield of **3d** was unusually lower than its analogues. So we monitored the first step for **3d** via LC-MS, which exhibited that the excellent selectivity of mono-substituted product **M5** to **BP11** is over 20:1. Meanwhile, the high polar interval of LC spectrum (0.2-1.8 min) displayed a broad mixture with m/z between 400 to 800, which was oligomers caused by the vinyl group lowering the yield of the first step. The polymerization can be accelerated via $B(C_6F_5)_3$. (Reference about polymerization accelerated by $B(C_6F_5)_3$: Yang, X., Stern, C. L. & Marks, T. J. Cation-like homogeneous olefin polymerization catalysts based upon zirconocene alkyls and tris(pentafluorophenyl)borane. *J. Am. Chem. Soc.* **113**, 3623–3625 (1991).)

Fig. A3.3 LC-MS of the first step for 3d

Q4: How easy they are to separate (in reaction)?

A4: Thank you for your inquiry. We separated the polysulfides easily with direct silica gel column chromatography acquiring with pure products (all characterized by NMR). The eluent composition of silica gel column chromatography and R_f value of TLC have been all added in experimental procedure in Supplementary Information.

Q5: We need to know the practicalities of this new method so as to be able to compare

it with other similar methods belonging to the methodology of double nucleophilic substitution of a Lg-S-S-Lg reagent.

A5: Thank you for your inquiry. We have compared our method with other similar methods. The results indicate that our methods possess irreplaceable advantage toward S_2Cl_2 or di(1-phthalimidyl)disulfane (**1b**).

1. Since S_2Cl_2 is a general reagent for tetrasulfide, we compared S_2Cl_2 with our reagent **1d** in tetrasulfide synthesis via 1H NMR and LC-MS. From the experimental results, S_2Cl_2 can't even achieve tetrasulfide synthesis with simple functional group in our work. Detailed information was added in Page S43-S47 of Supplementary Information.

1) Comparative reaction of **9a** synthesis between S_2Cl_2 and **1d** was carried out with detection of 1H NMR and LC-MS. 1H NMR of **9a** synthesis by S_2Cl_2 shows that the ratio of **9a** to **9a-S₅** is 2.5:1, much lower than 15:1 afforded by **1d**. UV detector of LC-MS exhibited that the ratio of **9a** to **9a-S₅** to **9a-S₃** given by S_2Cl_2 is 3.3:5:1, substantially different from 10:2:1 given by **1d**. Besides, there is a huge gap between the yields afforded by S_2Cl_2 and **1d** (8% vs 70%).

Scheme A5.1.1 Compared results of **9a** synthesis from S_2Cl_2 and **1d**

Fig. A5.1.1 1H NMR and LC-MS comparison of **9a** between S_2Cl_2 and **1d**

2) Another comparative reaction for **9h** synthesis between S_2Cl_2 and **1d** was carried out with detection of LC-MS. However, S_2Cl_2 could not afford any products except a series of polymers. Prominently, **9h** was afforded as the main product when reagent **1d** was applied in this reaction. S_2Cl_2 couldn't work out in tetrasulfide synthesis even with substrate possessing hydroxyl group.

Scheme A5.1.2 Comparison results for **9h** synthesis between S_2Cl_2 and **1d**

Fig. A5.1.2 LC-MS comparison of 9h between S_2Cl_2 and 1d

2. Di(1-phthalimidyl)disulfane (**1b**), which has been reported to be a disulfur transfer reagent in Harpp's work, was also subjected in unsymmetrical tetrasulfide synthesis. However, the selectivity and efficiency are rather poor. Only small amount of mono-substituted product (22%) was obtained with **1b**, which has been recorded in Entry 2 of Supplementary Table 3 before.

Scheme A5.2 Comparison results for 9h synthesis between S_2Cl_2 and 1d

The source data underlying **Figs. 1a-1e, 2a-2b, 3a-3b, 4, 5, 6, 7**, excel data of **Fig. 3b**, and excel data of **Supplementary Table 1-3** are provided as a Source Data file.

Thank you very much again for your assistance with our manuscript and helpful suggestion from three reviewers. We hope the above revisions will meet the requirements for publication on *Nature Communications*.

REVIEWERS' COMMENTS:

Reviewer #3 (Remarks to the Author):

I'm still concerned about the empirical objectivity of this work as reported. One can see that Professor Jiang's response letter admits to several complications and issues with his method as implied by the yields, and ones in line with my comments and requests from the last round. However, bar one comment about Harpp's reagents, none of these caveats has been summarised and brought into the new manuscript. Instead, Professor Jiang has decided to stick with his sales pitch, telling the readership just how good his new method is when professional, practising, synthetic organic chemists actually want to know about where the method falls down and why. As a result, I think he has missed a science opportunity. However, I rest my case, and that's it for me.

The detailed replies to reviewer's are listed as below:

Q1: I'm still concerned about the empirical objectivity of this work as reported. One can see that Professor Jiang's response letter admits to several complications and issues with his method as implied by the yields, and ones in line with my comments and requests from the last round. However, bar one comment about Harpp's reagents, none of these caveats has been summarised and brought into the new manuscript. Instead, Professor Jiang has decided to stick with his sales pitch, telling the readership just how good his new method is when professional, practising, synthetic organic chemists actually want to know about where the method falls down and why. As a result, I think he has missed a science opportunity. However, I rest my case, and that's it for me.

A1: There is no absolutely perfect thing in the world. Moreover, we have already demonstrated all the special examples originating from complex molecules with multiple functional groups, not on account of the reagent. For modern fast drug discovery, it is so important to acquire the complex design molecules. Please eradicate prejudice and evaluate objectively and impartially. Meanwhile, all the explanations have been added for the complex molecules in the revised manuscript.

1. We have added the detailed explanation about the low yield of **3d** in Page 4 of the Manuscript as below.

“Among them, compound **3d** was afforded with a yield of 55% with occurrence of polymerization of vinyl group accelerated by $B(C_6F_5)_3$ ³³.”

“33. Yang, X., Stern, C. L. & Marks, T. J. Cation-like homogeneous olefin polymerization catalysts based upon zirconocene alkyls and tris(pentafluorophenyl)borane. *J. Am. Chem. Soc.* **113**, 3623–3625 (1991).”

2. We have added the detailed explanation about the relative low efficiency of **3l**, **3m** and **3o** in Page 4 of the Manuscript as below.

“The relative low efficiency of **3l**, **3n** and **3o** with tryptophan motif resulted from a cyclic disulfide intermediate generated from nucleophilic cyclization of 2-position of indole (for details see the Supplementary Fig. 2).”

3. We have added the detailed explanation about the low yields of **6g** and **7h** in Page

8 of the Manuscript as below.

“Arylboronic acids derived from L-tyrosine and estrone were compatible in the cross-linkage, affording a pathway to late-stage modification of natural products (**6g** and **7h**), though the slow rate of transmetallation of boric acid with Cu[III] in the first step brought about lower efficiency.”

4. We have added the detailed description about the specific conditions of **8b**.

“(+)– δ -Tocopherol, a kind of vitamin E, could be disulfurated directly despite of the presence of free hydroxyl group (**8b**) under nitrogen atmosphere.”

5. We have added the description and figure about comparison with previous methods in Page 9 as below.

“S₂Cl₂, a common disulfur structure, hardly achieves multiple heteroatom ordered connection on account of its high activity and strong acidity. Taking synthesis of **9a** as an example, the selectivity of **9a** to **9a-S₅** is 2.5:1 when S₂Cl₂ was involved, much lower than 15:1 afforded by our reagent **1d**. Besides, there is a huge gap between the efficiency afforded by S₂Cl₂ and **1d** (8% vs 70%) (Fig. 8a). Di(1-phthalimidyl)disulfane (**1b**) reagent developed by Harpp³⁰, which avoids the disadvantage of acidity with S₂Cl₂, remains poor selectivity and low efficiency owing to the nondistinctive S-N bond. For instance, in the first step coupling with aniline, Harpp’s reagent gave a mono-coupling product in 30% and a bis-coupling in 60%. Optimistically, quantitative onefold **2f** could be obtained with our reagent **1f**.”

Fig. 8 | Comparison with previous reagents. **a** Comparison with S₂Cl₂. **b** Comparison with Harpp's reagent **1b**.